# Role of Non-Coding RNAs in Hepatocellular Carcinoma Progression: From Classic to Novel Clinicopathogenetic Implications

**DOI:** 10.3390/cancers15215178

**Published:** 2023-10-27

**Authors:** Mario Romeo, Marcello Dallio, Flavia Scognamiglio, Lorenzo Ventriglia, Marina Cipullo, Annachiara Coppola, Chiara Tammaro, Giuseppe Scafuro, Patrizia Iodice, Alessandro Federico

**Affiliations:** 1Hepatogastroenterology Division, Department of Precision Medicine, University of Campania “Luigi Vanvitelli”, Piazza Miraglia 2, 80138 Naples, Italy; mario.romeo@unicampania.it (M.R.); flavia.scognamiglio@unicampania.it (F.S.); lorenzo.ventriglia@unicampania.it (L.V.); marina.cipullo@studenti.unicampania.it (M.C.); annachiara.coppola@unicampania.it (A.C.); alessandro.federico@unicampania.it (A.F.); 2Biochemistry Division, Department of Precision Medicine, University of Campania “Luigi Vanvitelli”, Piazza Miraglia 2, 80138 Naples, Italy; chiara.tammaro@unicampania.it (C.T.); giuseppe.scafuro@unicampania.it (G.S.); 3Division of Medical Oncology, AORN Azienda dei Colli, Monaldi Hospital, Via Leonardo Bianchi, 80131 Naples, Italy

**Keywords:** non-coding RNAs, liver cancer, translational medicine

## Abstract

**Simple Summary:**

Given the increasing incidence of hepatocellular carcinoma (HCC) worldwide, a full comprehension of the molecular pathways supporting HCC onset and progression is urgently needed to design more tailored prognostic models and appropriate therapeutic approaches. A large number of pro-carcinogenic and anti-carcinogenic mechanisms are regulated by RNA non-coding transcripts known as non-coding RNAs (ncRNAs). This review reports the main dysregulated ncRNAs involved in HCC cancerogenesis and describes the relative implications on the progression mechanisms highlighting potential markers for the early diagnosis of HCC and targets for innovative therapeutic strategies.

**Abstract:**

Hepatocellular carcinoma (HCC) is a predominant malignancy with increasing incidences and mortalities worldwide. In Western countries, the progressive affirmation of Non-alcoholic Fatty Liver Disease (NAFLD) as the main chronic liver disorder in which HCC occurrence is appreciable even in non-cirrhotic stages, constitutes a real health emergency. In light of this, a further comprehension of molecular pathways supporting HCC onset and progression represents a current research challenge to achieve more tailored prognostic models and appropriate therapeutic approaches. RNA non-coding transcripts (ncRNAs) are involved in the regulation of several cancer-related processes, including HCC. When dysregulated, these molecules, conventionally classified as “small ncRNAs” (sncRNAs) and “long ncRNAs” (lncRNAs) have been reported to markedly influence HCC-related progression mechanisms. In this review, we describe the main dysregulated ncRNAs and the relative molecular pathways involved in HCC progression, analyzing their implications in certain etiologically related contexts, and their applicability in clinical practice as novel diagnostic, prognostic, and therapeutic tools. Finally, given the growing evidence supporting the immune system response, the oxidative stress-regulated mechanisms, and the gut microbiota composition as relevant emerging elements mutually influencing liver-cancerogenesis processes, we investigate the relationship of ncRNAs with this triad, shedding light on novel pathogenetic frontiers of HCC progression.

## 1. Introduction

Hepatocellular carcinoma (HCC) represents the most common primary liver cancer and a predominant malignancy with increasing incidences and mortalities worldwide [1]. In parallel with the chronic Hepatitis C virus (HCV), Hepatitis B virus (HBV) infection, alcohol consumption [Alcoholic Fatty Liver Disease (AFLD)], and aflatoxin B1 exposure [1], the progressive increase of Non-alcoholic Fatty Liver Disease (NAFLD) incidence has fueled HCC occurrence even in non-cirrhotic contexts [1,2]. In light of this, the full comprehension of the molecular pathways supporting HCC onset and progression is urgently needed to design more tailored prognostic models and appropriate therapeutic approaches.

A wide inter- and intra-individual molecular diversity, as well as the crucial role of genetics in the clinical outcome for HCC patients, has been reported [3]. The accumulation of several genetic (and epigenetic) alterations is considered the key driver of liver carcinogenesis, a multi-step process where the acquisition of a malignant phenotype represents only the beginning of a dramatic cascade of events [4].

For this purpose, a plethora of early gene-sequencing studies focused mainly on protein-coding genes, profiled the HCC-associated mutations, and subsequently defined the related pathobiological pathways, confirming a heterogeneous tumor microenvironment where various clusters of neoplastic cells showing different molecular gene signatures [3,5]. Hereafter, further findings have progressively shown that, at the hepatic level, as well as in other organs, a large number of pro-carcinogenic and anti-carcinogenic mechanisms are regulated also by RNA non-coding transcripts known as non-coding RNAs (ncRNAs) [6,7,8].

Conventionally, ncRNAs are classified in terms of their molecular size: ncRNAs shorter than 200 nucleotides are defined as small ncRNAs (sncRNAs), whereas longer ncRNAs are considered long ncRNAs (lncRNAs) [7]. However, beyond genetics, the systemic and local immune system status (both innate and acquired immune dysfunction [9]), and the gut microbiota composition (HCC-related dysbiosis “signature” [10]) represent equally relevant emerging, mutually influenced, and incompletely explored pathogenic frontiers composing the complex network of the still-not-fully clarified HCC pathogenesis.

In this review, by following the above-proposed classification based on molecular size, we report the main dysregulated ncRNAs involved in HCC cancerogenesis and describe the relative implications for progression mechanisms. Moreover, we explore ncRNAs as potential markers for the early diagnosis of HCC and targets for innovative therapeutic strategies.

## 2. Small Non-Coding RNAs and Hepatocellular Carcinoma

SncRNAs represent a heterogeneous group of gene expression regulators embracing different classes of small (shorter than 200 nucleotides) molecules including, among several others [PIWI-interacting RNAs (piRNAs), transfer RNAs (tRNAs), small nucleolar RNA (snoRNA), and small nuclear RNA (snRNA), tRNA-derived-small RNAs (tsRNAs), vault RNA-derived-small RNAs (vtRNAs), and Y RNA-derived-small RNAs (ytRNAs)], microRNAs (miRNAs), and small interfering RNAs (siRNAs), whose role in biological processes fueling human carcinogenesis has been largely documented [11].

MiRNAs represent endogenous transcripts and the most investigated class of sncRNAs involved in post-transcriptional gene expression regulation in eukaryotes [11]. Conversely, despite not being primarily expressed in mammals, the small interfering RNAs (siRNAs) constitute another class of exogenous sncRNAs very similar to the miRNAs in terms of biological function and are involved equally in the post-transcriptional RNA interference (RNAi) pathways [7,12].

In the HCC context, given the predominant pathogenetic role and the nature of endogenous molecules, we herein focus on reporting the miRNA dysregulation affecting molecular pathways driving hepatic cancer progression, reserving for siRNAs, in light of their potential ab extrinsic guided action on specific targets, a subsequent section, specifically dedicated to the implications of ncRNAs in the treatment of this neoplasm.

### 2.1. Role of microRNA Dysregulation in Hepatocellular Carcinoma Progression

In cancer-related pathogenetic contexts, on the basis of specific post-transcriptionally inhibited target genes, miRNAs are conventionally distinguished as oncomiRNAs (“oncomiRs”), contributing to the cancer onset and progression, and tumor-suppressor miRNAs (“TS-miRs”), which pleiotropically mediate the antitumor response [13,14]. The downregulation of TS-miRs and the upregulation of oncomiRs represent the crucial events even in HCC development and worsening as well as the epiphenomenon of different molecular aberrations that may depend on genetic alterations, epigenetic modifications affecting DNA histones, transcriptional misregulation, and damage of the processing machinery genes involved in miRNA synthesis.

Several miRNAs, exerting physiologically important regulatory functions in the liver, were found to be dysregulated in HCC. In this complex picture, specific miRNAs’ dysregulation represents a constant pathogenetic feature driving all the clinical HCC stages, from the earliest to the most advanced.

The next subparagraph gives a general overview of the major TS-miRNAs and oncomiRs transversally dysregulated in different chronic hepatopathies; afterward, in the subsequent subsections, the dysregulation of miRNAs in specific chronic liver disease contexts contributing to HCC progression, with a view to the pathogenetic-related mechanisms, is reported.

#### 2.1.1. Principal TS-miRNAs in Hepatocellular Carcinoma Progression

Although the comprehension of the mechanisms specifically related to the etiology of chronic liver disease is of crucial importance, some molecular pathways appear regulated by shared ncRNAs transversally uniting various causes of chronic liver diseases.

In line with this, reduced expression levels of *miR-122* have been found both in NAFLD patients and in a subset of HCC patients including HBV-positive individuals with highly invasive and metastatic cancer [15] Of relevance, this molecule appeared able to attenuate HCC progression both in NAFLD and HBV contexts. In support of this, *miR-122* genetically depleted mice progressively develop steatohepatitis, fibrosis, and HCC, establishing it as a bona fide TS; conversely, in animal models, the delivery of *miR-122* (by using a viral vector or liposomal nanoparticles) resulted in liver tumor suppression [15]. These results, intending to translate the basic research to the bedside, revealed the crucial need to develop clinical-routine-appliable strategies based on the direct monitoring of *miR-122* in biological fluids as well as on *miR-122* supplementation in NASH or HBV-positive HCC patients.

*MiR-29* is another TS-miRNA involved in the regulation of oncogenetic processes (proliferation, neoangiogenesis, and metastasis) in different human neoplasms, including viral- and not-viral-related HCC. Regarding this, *miR-29* has been reported to contrast HCC progression via specifically targeting and downregulating the Insulin-like growth factor 2 mRNA binding protein 1 (IGF2BP1) and Vascular Endothelial Growth Factor A (VEGFA), as well as anti-apoptotic proteins such as B-cell lymphoma 2 (BCL2) [16,17,18].

Similarly, *miR-195* has been shown to impede angiogenesis by targeting proteins such as VEGF, Vav guanine-nucleotide exchange factor 2 (VAV2), and cell division cycle 42 (CDC42), ubiquitously involved in cancer progression, including HCC [19].

In general, regardless of the underlying etiology, neoangiogenesis coupled with the cell’s acquisition of invasiveness properties represent two crucial moments in HCC metastasizing. In this context, RHO-associated protein kinase (ROCK) is considered a pivotal downstream target of the RhoA GTPase pathway deputed to the regulations of actomyosin bundles and focal adhesions that normally impede epithelial–mesenchymal transition, angiogenesis, and thus, metastasis [20]. When constantly overexpressed, in the HCC context, ROCK confers cell motility and invasiveness properties. In this scenario, *miR-101* has been shown to target ROCK, appearing as a crucial TS-miR whose down-regulation may represent a crucial pathogenetic event in determining extra-hepatic HCC spreading [21]. In the same vein, in a recent study, Zhang and coworkers revealed *miR-497* as an important TS-miR demonstrating the capability of this molecule to contrast proliferation, invasion, and metastasis by targeting the Rictor/AKT pathway in hepatoma cells [22].

#### 2.1.2. Principal oncomiRNAs in Hepatocellular Carcinoma Progression

In the last few decades, a constantly growing number of findings have revealed the overexpression of various miRNAs in HCC of various etiologies and progressively clarified the relative targets, suggesting the role of these molecules as oncomiRs in liver cancerogenesis [23].

Among these, *miR-221* represents one of the most investigated and highly expressed miRNAs in HCC tissues. Interestingly, *miR-221* has been associated with proliferation and apoptosis dysregulation in HCC contexts. In particular, in animal models, the overexpression of this molecule promotes the growth of tumorigenic murine hepatic progenitor cells targeting the DNA damage-inducible transcript 4 (DDIT4)/mTOR pathway [24]. In addition, the revelation of its capability to interfere with apoptosis by targeting tumor suppressors such as Phosphatase and tensin homolog (PTEN) and Metallopeptidase Inhibitor 3 (TIMP3) via the activation of the AKT pathway further confirms its oncomiR nature [25].

*MiR-21* represents a relevant oncomiR playing a key pathogenetic role in different liver diseases, including viral hepatitis, NAFLD, and AFLD. In these contexts, the overexpression of this molecule, by activating Hepatic Stellate Cells (HSCs), favors collagen synthesis, fibrosis, and thus, hepatocarcinogenesis [26]. In vitro, *miR-21*, by targeting the tumor suppressor Kruppel-like factor 5 (KLF5), promoted hepatic cancer cells’ invasiveness, as well as cancer progression, by inducing HCC cells to secrete angiogenic mediators, including VEGF [27]. In humans, consistent with all these findings, the overexpression of this miRNA, in both cancer tissues and serum, was largely reported in patients with HCC and, interestingly, the expression levels appeared significantly correlated with tumor progression [28,29,30]. In the same vein, in a recent comprehensive study, Sathipati et al. highlighted the importance of miRNA signatures in predicting the HCC progression through the analysis of 348 expression profiles of 540 miRNAs derived from 348 HCC patients’ data reported in “The Cancer Genome Atlas” (TCGA) database [31]. Specifically, the dataset included 258 patients affected by early-stage disease and 90 patients with advanced HCC. Interestingly, the analysis revealed the expression of 7 [homo sapiens (hsa)-miR-550a, hsa-miR-424, hsa-miR-574, hsa-miR-512, hsa-let-7i, hsa-miR-549, and hsa-miR-518] of the 10 top-ranked advanced-associated-HCC miRNAs significantly associated with overall survival [31]. Pragmatically considering the pathogenetic implications of these emerging molecules: the role of has-miR-549 in human biological processes appears still obscure; the targets of hsa-let-7i, with the exception of genes involved in fibrosis process regulation, have been well elucidated in other neoplasms such as the metastatic colorectal cancer [32], but remain not completely defined in HCC, as have the *modus agendi* of hsa-miR-518, whose cruciality in chronic inflammatory pathologies of extrahepatic organs has, however, been widely described [33].

Conversely, the centrality of miR-550a, miR-424, miR-574, and miR-512 in promoting HCC progression has been widely highlighted.

In particular, *miR-550a-5p* was shown to promote the proliferation and migration of HCC both in vitro and in vivo in a xenograft tumor model by targeting glucosamine (UDP-N-acetyl)-2-epimerase/N-acetylmannosamine kinase (GNE) via the Wnt/β-catenin signaling pathway. GNE is a bifunctional enzyme involved in the regulation of the metabolic pathway responsible for the N-acetylneuraminic acid (NeuAc) biosynthesis, a precursor of sialic acids [34]. Sialic-acid-induced modifications of cell surface molecules are crucial for their role in many biological processes, including cell adhesion and signal transduction [34]. Indeed, differential sialylation is involved in the tumorigenicity and metastatic behavior of malignant cells via the dysregulation of the Wnt/β-catenin signaling axis [35,36]. Accordingly, *miR-550a-5p* was shown to regulate the progression of HCC, being associated with poor prognosis [37].

In the same vein, very recently Feng et al. showed the role of *miR-424-3p* in the transition to metastatic HCC by targeting the Serum Response Factor (SRF)—STAT1/2 axis [38]. In particular, *miR-424-3p* exhibited a strong association with HCC cell migration and invasion in vitro, contributing also to metastasis development in vivo [38]. SRF is a direct functional target of *miR-424-3p* and is required for its oncogenic activity, being a transcription factor involved in cell cycle regulation, apoptosis, cell growth, and differentiation. Overall, miR-424-3p reduces the interferon pathway by attenuating the transactivation of SRF on STAT1/2 and Interferon Regulatory Factor 9 (IRF9) genes, which, in turn, enhances the matrix metalloproteinases (MMPs)-mediated extracellular matrix (ECM) remodeling [38].

Notably, liver-cancer-cell-secreted exosomes promote bone metastasis derived from primary liver cancer facilitating osteoclast differentiation through the *miR-574–5p*/Bone morphogenetic protein 2 (BMP2) axis [39]. BMP2 belongs to the Transforming growth factor (TGF)-β proteins superfamily and was shown to be a direct target of *miR-574–5p* mediated osteoclastogenesis [39]. In support of this, a previous study conducted by Tessitore et al. underlined the role of miR-574–5p in the progression of high-fat diet (HFD)-induced NAFLD-Non-alcoholic steatohepatitis (NASH)-HCC in C57BL/6J mice supporting the involvement of this miRNA in HCC progression in vivo [40].

Another recent finding by de la Cruz-Ojeda et al. showed the importance of *miR-512-3p* as a biomarker signature of advanced HCC and Sorafenib efficacy in this clinical setting [41]. In particular, the pro-tumoral role of time-dependent circulating levels of miR-512-5p was demonstrated in two independent cohorts of advanced [Barcelona Clinic Liver Cancer (BCLC)-C]-stage HCC patients and further confirmed by the increased expression in HCC tissues compared with adjacent nontumor samples obtained by liver resection [41]. Moreover, Sorafenib treatment induced miR-512-3p downregulation in the HepG2 cell line, reducing cell proliferation and migration. Conversely, functional studies showed that the administration of the *miR-512-3p* mimic increased the cells’ aggressiveness in the neoplastic sense. *MiR-512-3p* also controlled oxidative balance by blocking mitochondrial metabolism, mitochondrial complex assembly, and, consequently, oxidant response through the downregulation of several mito-ribosomal proteins (MRPS) such as MRPS22 or MRPS36 [41].

Figure 1 schematizes the main oncomiRs and TS-miRs, the related targets, and the relative principal implications (proliferation, angiogenesis, metastasis) in the progression of HCC.

However, at this point, it appears also necessary to take into account the background that characterizes HCC development considering the different aetiologies and the related dysregulated mechanisms, as specifically reported in the next subparagraph.

In the HCC microenvironment, various overexpressed miRNAs favor tumor progression; in particular, miR-221 [targeting TIMP1/PTEN (AkT), and DDIT4 (mTOR) pathways] [25], miR-302d (targeting TGFBR2) [42], miR-96 (targeting Caspase-9) [43], miR-25 (targeting TRAIL) [44], and miR-92a (targeting FBXW7) [45] impact on cancer cell proliferation and survival mechanisms, whereas miR-130 (targeting HOXA5) [46] and miR-210 (targeting FGFRL1) [47] are principally involved in promoting angiogenesis. The overexpression of miR-574 (targeting BMP2) [39], miR-214 (targeting WASL) [48], miR-424 (targeting the SRF—STAT1/2 pathway) [38], and miR-135 (targeting FOXO1) [49] facilitates invasion, migration, and metastasization. Moreover, miR-18a (targeting Bcl2l10) [50], miR-550a (targeting the GNE—β-catenin/Wnt pathway) [37], and miR-512 (targeting various MRPS) [51] have been reported to influence both proliferation/apoptosis and invasion/migration/metastasization processes, as well as miR-21 (targeting the hKLF5-PI3K-Akt pathway) [29], miR-155 (targeting PTEN) [52], and miR-873 (targeting TSLC1, among others) [53], whose pleiotropic involvement in various crucial cancer-progression mechanisms (proliferation, angiogenesis, invasion, and metastasis) has been widely reported. In HCC cells, the downregulation of miR-195 (targeting CDC42 and other proteins regulating the cell cycle) [19], miR-15b (targeting Bcl2) [54], miR-26a (targeting ULK1) [55], miR-766 and miR-148 (influencing the β-catenin/Wnt pathway) [56,57] represents dramatic events for the loss of control of proliferation/survival mechanisms. Moreover, miR-126 (targeting VEGF) [58], miR-451 (targeting IL6R) [59], and miR-338-3p (targeting VEGF and MACC1) [60] are pivotally involved in HCC angiogenesis. The downregulation of miR-539 (targeting FSCN1) [61], miR-101 (targeting ROCK) [22], miR-497 (targeting Rictor) [22], miR-345 (targeting INF1) [62], and miR-200a (targeting GAB1) [63] favor invasion, migration, and metastasization, consistently with the following miRNAs which also regulate HCC-related proliferation/apoptosis processes: miR-31-5p (targeting SP1) [64], miR-33b (targeting SALL4) [65], and miR-29a (targeting CLDN1) [66]. With respect to the latter, some evidence supports miR-29 (targeting, among several others, BCL2, IGF2BP1, VEGF) [16,17,18] involvement in each of the crucial cancer-progression mechanisms as well as for miR-122 (targeting, among others, p53 and MDM2) [67] and miR-1301 (targeting, among others, BCL9, VEGF, and β-catenin pathway) [68]. TIMP1: TIMP metallopeptidase inhibitor 1; PTEN: Phosphatase and TENsin homolog; DDIT4: DNA damage-inducible transcript 4; mTOR: mammalian target of rapamycin; TGFBR2; transforming growth factor beta receptor 2; TRAIL: Tumor-necrosis-factor-related apoptosis-inducing ligand; HOX: Homeobox; FGFRL1: Fibroblast growth factor receptor-like 1; BCL: B-cell lymphoma; GNE: UDP-N-acetylglucosamine 2-epimerase/N-acetylmannosamine kinase; WASL: WASP-like actin nucleation promoting factor; MRPs: Mitochondrial ribosomal proteins; KLF5: Krüppel-like factor 5; TSLC1: tumor suppressor of lung cancer 1; ULK1: Unc-51-Like Autophagy Activating Kinase 1; MACC1: Metastasis-associated in colon cancer 1; SALL4: Sal-like protein 4; Insulin-like growth factor 2 mRNA-binding protein 1.

### 2.2. Etiology-Specific Dysregulated miRNAs in Hepatocellular Carcinoma Progression

As a golden rule, the etiology determining chronic liver disease represents the driving force for HCC progression [69,70]. Strictly dependent on this, especially focusing on the regulation employed by miRNAs, numerous molecular abnormalities were found in specific etiologic contexts. The most common cause of HCC remains liver cirrhosis (LC), which generally represents the end stage of various chronic hepatopathies, with a predominant reference to NAFLD, AFLD, and chronic HBV and HCV infections [70,71].

Regarding dysmetabolic surroundings, various pieces of evidence suggest the potential role of *miR-155* as an oncoMir favoring the early stages of hepatocarcinogenesis in NAFLD contexts [72,73]. In brilliant research by Wang and coworkers, the choline-deficient and amino-acid-defined (CDAA) diet of 84 weeks, a regimen able to promote nonalcoholic steatohepatitis (NASH)-induced hepatocarcinogenesis, was associated with a significant upregulation of miR-155 in C57BL/6 mice, contemporarily with a reduced expression of its natural target CCAAT/enhancer binding protein beta (C/EBPbeta), a key transcription factor regulating the proliferation and differentiation of multiple cell types [74]. Consistently, in the same study, miR-155 was also significantly upregulated in primary human HCCs with a simultaneous decrease in C/EBPbeta levels compared with matching hepatic tissues [74].

In a very recent study, Niture and coworkers shed light on the role of *miR-483-5p* in HCC progression and its biological significance as a linking factor between NAFLD, AFLD, and HCC both in vitro and in vivo [75]. The downregulation of *miR-483-5p* expression was proven in 40 HCC samples compared to 30 adjacent normal tissues and 10 adjacent cirrhotic tissues and was strongly associated with Notch 3 upregulation [75,76]. Conversely, the overexpression of *miR-483-5p* in HepaRG and HCC cells dysregulated Notch signaling, inhibiting cell proliferation/migration, induced apoptosis, and increased sensitivity towards the antineoplastic agents Sorafenib/Regorafenib [75,76]. Appealingly, the inactivation of *miR-483-5p* increased cell steatosis and fibrosis by modulating lipogenic and pro-fibrotic gene expression. Mechanistically, *miR-483-5p* targets Peroxisome-proliferator-activated receptor alpha (PPARα) and Tissue inhibitor of metalloproteinases 2 (TIMP2) gene expression leading to the suppression of steatosis and fibrosis. PPARα regulates lipid metabolism, whilst TIMP2 is a tissue inhibitor of the metalloproteinase family whose dysregulation has been associated with the progression of various tumors [77,78]. Additionally, the downregulation of *miR-483-5p* was observed in the liver of HFD-fed mice or a standard Lieber–Decarli liquid diet containing 5% alcohol, leading to increased hepatic steatosis and fibrosis [75]. Overall, *miR-483-5p* was shown to play a critical role in the inhibition of cell steatosis and fibrogenic signaling as a TS in HCC, representing a potential therapeutic target for the management of NAFLD-related cancer.

Along with metabolic pathway abnormalities, it is important to also consider the strong impact of viral infections in the dysregulation of pivotal pathways responsible for the onset and progression of HCC [79,80].

Qin and coworkers demonstrated the role of hepatitis B virus surface antigen (HBsAg) in the induction of stemness of HCC through the regulation of *miR-203a* [81]. Specifically, *miR-203a* was found to be significantly downregulated in HBsAg-positive HCC patients in microarray analysis performed on 55 frozen liver tumors and normal adjacent tissues, as well as 4 fresh tumor tissues [81].

Accordingly, this prompted the investigation of the regulatory function both in vitro and in vivo through functional experiments [81]. Indeed, the negative correlation between *miR-203a* and HBsAg expression was further confirmed by performing quantitative real-time polymerase chain reaction (PCR) after stimulation or overexpression/knockdown of HBsAg in vitro. Furthermore, the biological attitude of *miR-203a* to act as a tumor suppressor inhibiting proliferation, migration, and clonogenicity was proved through its overexpression in xenograft mice models. Even more, the increased expression of miR-203a remarkably increased the sensitivity to 5-fluorouracil (FU)—treatment inducing a decrease in the proportion of HCC cells showing stem markers. miR-203a molecularly targeted the oncogene “B lymphoma Mo-MLV insertion region 1 homolog” (BMI1 gene) responsible for the maintenance of the self-renewal capacity of stem cells [82].

Consistently, a significant negative correlation between *miR-203a* and BMI1 was shown in the HCC specimen and opposite expression in BMI1 was observed after overexpression/knockdown of *miR-203a* in vitro [81].

Ultimately, given the noteworthy role of miRNAs in the onset and progression of HCC, the following section of the review explores the recent data published on the molecular mechanisms regulated by miRNAs in HCC progression, paying attention as well to the recent findings concerning specific aetiologies of HCC.

Figure 2 synthetizes the main dysregulated miRNAs and the relative regulated molecular pathways in specific etiological (AFLD/NAFLD and chronic viral hepatitis) contexts.

The overexpression of miR-550a-5p downregulates glucosamine (UDP-N-acetyl)-2-epimerase/N-acetylmannosamine kinase (GNE) and consequently N-acetylneuraminic acid (NeuAc) (dashed black arrow) leading to hepatocytes modifications of cell surface molecules which induce tumoral behavior through the dysregulation of the Wnt/β-catenin signaling axis (solid black arrows). Overexpression of miR-424-3p is associated with migration and invasion of HCC cells in vitro and the development of metastasis in vivo targeting the Serum Response Factor (SRF) involved in cell cycle regulation, apoptosis, cell growth, cell differentiation, and STAT1/2 axis activation (green arrow). The overexpression of miR-574-5p downregulates Bone morphogenetic protein 2 (BMP2) facilitating osteoclast differentiation and inducing bone metastasis of liver primary cancer (blue arrow).

The overexpression of miR-512-3p is associated with more aggressive behavior of HCC cells by blocking mitochondrial metabolism, mitochondrial complex assembly, and consequently, the oxidant response through the downregulation of several mito-ribosomal proteins such as MRPS22 or MRPS36 (red arrow). Downregulation of miR-483-5p links HCC, NAFLD, and AFLD, inducing the upregulation of PPARα and TIMP2 responsible for the increased steatosis and fibrosis and HCC progression as well (brown arrow). Downregulation of miR-203a induced by HbsAg results in the low expression of BMI1, which is involved in maintaining the self-renewal capacity of stem cells (yellow arrow).

## 3. Long Non-Coding RNAs and Hepatocellular Carcinoma

Unlike sncRNAs, linear lncRNAs are transcripts longer than 200 nucleotides and share several common features with mRNAs, such as the transcription by RNA polymerase II, the 5′capping, the splicing, and the polyadenylation. lncRNAs are less expressed than mRNAs and they have tissue and developmental-stage-specific expression profiles [83]. In detail, lncRNAs constitute a heterogeneous large group of ncRNAs subdivisible, according to their genomic biogenesis, into five categories: (a) *intronic lncRNAs*, which originate from intronic regions within protein-coding genes; (b) *intergenic lncRNAs*, whose transcription occurs at the intergenic region level, between two protein-coding genes; (c) *enhancer lncRNAs*, whose transcription occurs in the enhancer sites of the genome; (d) *bidirectional lncRNAs*, which are transcribed bidirectionally in the promoter-regions of protein-coding genes; and (e) *antisense lncRNAs*, which originate from overlapping protein-coding genes and from which they are transcribed following the antisense direction [83]. Consistently with their heterogeneous biogenesis, the linear lncRNAs, whose heterogeneity appears to mirror their crucial regulatory functions in cancerogenesis processes, are very heterogeneous [83]. In this sense, linear lncRNAs can guide, after selectively binding, specific proteins on the cellular surface, and drive the release of the substance in the opportune target cell (“guide lncRNAs”). Moreover, they can act as molecular decoys to bind and facilitate the degradation of targeted proteins or other RNAs (“decoy lncRNAs”). Furthermore, they can serve as scaffolds (“scaffold lncRNAs”) functioning as a pivotal platform to assemble different molecular components (other RNAs and proteins). Lastly, by interacting with transcription factors or chromatin-modifying enzymes, they can tissue-specifically regulate the expression of targeted genes [83].

A particular emerging class of ncRNAs with a specific morphology is represented by the circular RNAs (CircRNAs), covalently closed RNA molecules modulating gene expression by acting as transcriptional regulators, miRNA sponges, and protein templates [84]. Due to their pleiotropic modes of action, circRNAs have been proposed as diagnostic and prognostic HCC markers and their potential role is reported in detail in the dedicated subsection of this paragraph.

### 3.1. Role of Linear Long Non-Coding RNA Dysregulation in Hepatocellular Carcinoma Progression

In recent years, it has been shown that lncRNAs can modulate different stages of liver diseases, affecting immune responses, liver regeneration, and redox signaling [85,86].

The dysregulation of lncRNA patterns promotes the worsening of liver pathologies, liver outgrowth, and oxidative stress, which eventually result in the initiation and progression of HCC through different signaling pathways [86]. Multiple lncRNA profiling studies on whole-genome transcriptome sequencing platforms have highlighted the altered expression pattern of lncRNAs in human HCC, suggesting a significant difference in the global expression profile between tumors and nontumor cells [86].

Abnormal cell proliferation is identified by cell cycle disturbances that result in uncontrolled cell division.

lncRNAs are involved in different phases of the cell cycle by directly and indirectly influencing the activities of several Cyclin-dependent kinases (CDK) [87].

For instance, the interaction of *Lnc-UCID* with DHX9 (DExH-Box Helicase 9) determines CDK6 enhanced expression, the promotion of G0/G1 to S phase transition, and, ultimately, HCC cell proliferation [88].

A central regulator of cell cycle progression is the E2F transcription factor family, whose expression and functions are tightly linked with the occurrence and development of various malignant tumors [89]. In this regard, many lncRNAs exhibit involvement in HCC progression due to the regulation of E2F family members, like E2F1 and E2F2. For instance, lncRNA *CASC11* enhances the stability of E2F1 mRNA by recruiting EIF4A3, which leads to the upregulation of E2F1 and, ultimately, promotes hepatocarcinogenesis and sponging mir-296–5p in HCC cells. It consequently activates the Wnt/β-catenin pathway by enhancing SOX12 and promoting HCC cell proliferation and progression [90].

Among scaffold lncRNAs, *HULC lncRNA* has been well-characterized as an oncogenic molecule, singularly upregulated in human HCC [91]. *HULC lncRNA* is implicated in HCC progression mainly by functioning as a molecular decoy of miR-107, which usually represses the E2F1 transcription factor [92]. Lu and colleagues demonstrated HULC’s capability to disable miR-107 functioning in the HCC microenvironment and, consequently, promote the activation of spongiosine kinase 1 with the stimulation of angiogenesis metastatic cell dissemination processes [92]. Consistent with this, Zhu and colleagues reported the capability of HULC to downregulate miR-29, whose TS-miRNA properties have already been described in Section 2.1.1 above [93].

*HOTAIR* displays its role in hepatoma cells at the epigenetic level. Indeed, it was shown to be able to interact and guide polycomb group complex 2 (PRC2) in the repression of specific target genes via the installation of transcriptional repressive histone 3 lysine 27 trimethylation (H3K27me3) [94]. It has been described by Fu and others that *HOTAIR* epigenetically silences miR-218-2 on chromosome 5, with the intervention of PRC2 to suppress P14 and P16 (tumor suppressor genes) signaling in the HepG2 cell line [95].

### 3.2. Etiology-Specific Dysregulated Linear lncRNAs in Hepatocellular Carcinoma Progression

Recently, several findings supporting the implication of lncRNAs in the pathogenesis of NAFLD-related HCC were published.

As mentioned in Section 2.2 above, the dysregulation of miR-155 expression represents a relevant pathogenetic moment, contributing to the onset of HCC in the early stage of cancerogenesis in the NAFLD context [72,73]. In line with this, Yuan et al. revealed miR-155 as a target of the lncRNA *CASC2*: in particular, *CASC2* was found to be downregulated in HCC tissue samples and was revealed to act as a molecular sponge inhibiting the expression of miR-155 [96].

Moreover, Wang et al. reported that the overexpression of lncRNA *NEAT1* was positively correlated with the upregulation of acetyl-CoA carboxylase (ACC) and fatty acid synthase (FAS), both involved in NAFLD pathogenesis [97]. Conversely, treatment with si-*NEAT1* lentivirus reduced the triglyceride and cholesterol levels in the animal model, further supporting the involvement of *NEAT1* in NAFLD progression. Additionally, in 2019, Si-si Jin and colleagues demonstrated how *NEAT1* affects miR-506 expression in BRL3A cells treated with free fatty acids (FFA) [98]. Hence, *NEAT1* was upregulated while miR-506 was downregulated in the progression of NAFLD; furthermore, *NEAT1* and miR-506 were proven to regulate fibrosis, inflammatory response, and lipid metabolism [98].

Another lncRNA that seems to exert a critical role in NAFLD-related HCC is *MALAT1*, whose expression was found to be elevated in liver tissue of NAFLD patients and HepG2 cells treated with 1 mM of FFA as an NAFLD in vitro model [99]. The knockdown of *MALAT1* upregulated the expression of PPAR and reduced CD36 levels, thus reversing fatty-acid-induced lipid accumulation in HepG2 cells [99]. *MALAT1* is also a crucial oncogene involved in the upregulation of Serine/arginine-rich splicing factor 1 (SRSF1) and the activation of the Wnt pathway, thus promoting HCC growth and development in liver tumors of the HCC mouse model [100]. Altogether, these findings highlighted a possible involvement of *MALAT1* as a new potential therapeutic target.

LncRNAs also exhibit a crucial role in infection-related HCC.

For instance, the integration of HBV into a normally silenced region of chromosome 8p11.21 produces a novel chimeric *HBx-LINE1* lncRNA, which exerts its function HBx-LINE1 by activating the Wnt signaling pathway and promoting hepatic injury by acting as a decoy to sequester liver-specific miR-122 [101]. Finally, high levels of *HBx-LINE1* have been found in HBV-related HCCs, showing the importance of *HBx-LINE1* as an independent prognostic factor in this context [101]. Moreover, HBV protein X (HBx) is highly carcinogenic, and 90% of HBx transgenic mice develop HCC [102]. HBx proteins seem to be implicated in the activation of several lncRNAs in HBV-related HCCs, such as DBH-AS1, lncRNA-UCA1, and HULC in HCC cell lines [103].

A profiling expression study by Zhang et al. demonstrated an altered lncRNA expression in a series of patients with different stages of HCV-associated liver disease [104]. *Linc01419* was found to be highly expressed in early-stage HCCs compared with pre-malignant dysplastic nodules [104]. In addition, another lncRNA, *K021443*, was found to be overexpressed in advanced HCC [104].

Additionally, *Linc01152* overexpression determines an increased HCC cell proliferation and tumor formation in nude mice, probably due to the activation of the transcription of IL-23 (interleukin 23), which induces the STAT3 pathway. Strikingly, linc01152 is described as being downregulated in all viral-hepatitis-related HCC through an unknown mechanism [104,105].

### 3.3. Circular RNAs in Liver Cancer Progression: Novel Diagnostic/Prognostic Markers?

In the following parallel-natured pathogenetic–clinical section, by focusing particularly on circRNAs correlated with HCC genesis and progression, we present an overview of ncRNAs that represent potentially useful novel tools in the diagnosis of HCC, as well as ncRNAs that can predict individual outcomes. In the last decade, also considering their well-demonstrated key regulator role in cancer-related molecular pathways, the world of circRNAs has been deeply investigated. In terms of downregulation or overexpression, a growing number of emerging studies provide evidence of several dysregulated circRNAs creating a complex pathomolecular network in the HCC context.

Preliminary observational research revealed higher has_circ_0005075 expression levels in HCC tissues compared with corresponding nontumorous counterparts. In addition, has_circ_0005075 expression was significantly upregulated in tumoral specimens, and, relevantly, the correlation analysis including several clinicopathological parameters of HCC patients demonstrated the relationship between has_circ_0005075 expression levels and HCC tumor size. However, despite these interesting results, its function remains to be clarified [106]. For this purpose, a subsequent brilliant in vitro study by Li et al. showed the association of has_circ_0005075 HCC overexpression with increased numbers of proliferative, migrated, and invasive SMMC-7721 cells, as well as the capability of this circRNA to promote HCC progression through the downregulation of miR-431 [107].

Consistent with this, Pan et al., by evaluating miR-431 expression levels in 95 HCC cases, highlighted that this miRNA was markedly downregulated in the HCC samples compared with corresponding adjacent nontumoral tissues [108].

However, even though the correlation analysis provided evidence for the relationship of miR-431 downexpression with multiple malignant characteristics, including lymph node metastasis, the HCC diagnostic accuracy of the assessment of miR-431 expression levels appeared not likewise elevated [Area under the Receiving Operator (ROC) Curve (AUC): 0.66] [108].

Moreover, in a recent in vitro–in vivo study, Sun et al. brilliantly investigated the function and regulation of circ_0038718 in HCC, suggesting its crucial role in fueling liver cancer progression [109]. The authors highlighted high expression levels of circ_0038718 in cell lines and HCC specimens and correlated this result with worsened prognosis in HCC patients. In vitro, circ_0038718 knockdown reduced HCC advancement mechanisms as cancer cell proliferation and cellular invasion [109]. Mechanistically, this molecule may act as the sponge of TS miR-139-3p, and, in support of this, the inhibition of miR-139-3p abrogated the regulatory effect of circ-0038718 in HCC cells, suggesting miR139 as a circ_0038718-specific target [109]. In the same vein, a group of researchers had previously tried to identify differentially expressed human miRNAs between HCC and normal liver tissues by evaluating the miRNA expression profiles of 375 HCC and 50 normal liver tissues. The authors shed light on has-miR-139-5p as a potential discriminant marker, and, further, the ROC curve analysis suggested that the survival prediction model developed based on tumor stage and hsa-miR-139-5p expression levels exhibited good performance in predicting the 3-year overall survival (OS) of HCC patients. By combining these findings, the encouraging results proposed circ_0038718 and miR-139 as crucial regulators of liver cancer progression and promising reciprocally influenced prognostic indicators, while also providing potential therapeutic targets for HCC treatment [97,109].

Despite the progressively increasing evidence on circRNA- and miRNA-related implicated in the progression of HCC, some circRNAs continue to be “orphans” of specific miRNAs, and vice versa, suggesting that several molecular mechanisms have yet to be elucidated. For instance, although the TS miR-122 represents the most frequently detected miRNA in the liver and its crucial role in cancer suppression has been largely demonstrated, circRNAs specifically targeting and regulating this molecule expression and functioning have not been identified [15]. Of relevance, miR-122 appears also closely related to the prognosis of HCC: decreased miR-122 levels have been associated with cell proliferation, invasion, and metastasis mechanisms, and several targets of miR-122 have been implicated in tumorigenesis, including ADAM10, cyclin G1, SRF, Wnt1, and IGF1R [110]. Conversely, circ_0128298 represents a circRNA orphan for relative-targeted miRNAs. Due to its capability of promoting proliferation and metastasis by not-fully clarified molecular pathways, and considering its significant upregulation in liver cancer samples compared with those of peritumorous tissues, this molecule has been proposed as another relevant circRNA associated with HCC progression [111]. If on one side, it has been demonstrated to have only a moderate HCC diagnostic accuracy [AUC: 0.66; sensitivity: 0.71; specificity: 0.81], on the other, this circRNA tissue expression relevantly correlated with the OS [111].

However, the current goal in this context appears to approach circRNA biological effects by identifying specific circRNA signatures. In this sense, a recently published meta-analysis including a total of eight studies highlighted, through adequate ROC curve analysis, an elevated diagnostic accuracy [AUC: 0.86; sensitivity: 0.78; specificity:0.80] for the specific circRNAs expression profile [downregulated circRNAs: hsa_circ_0003570, circZKSCAN1, hsa_circ_0001649, hsa_circ_0004018 tissue levels, and hsa_circ_0001445 serum levels; overexpressed circRNAs: hsa_circ_0005075, hsa_circ_0091582, and hsa_circ_0128298 tissue levels] in confirming HCC [112]. In addition to the potential diagnostic role, survival analyses also revealed a relevant prognostic implication by showing that abnormally expressed circRNAs were intimately associated with tumor size, serum alpha fetal protein level, differentiation grade, microvascular invasion, and metastasis in patients with HCC, as well as that the downregulated circRNA expression signature correlated perfectly with HCC survival [hazard ratio (HR): 0.42], whereas the HCC cases with high circRNA levels had significantly poorer prognoses than those of patients with low circRNA levels (HR: 2.22). In the same vein, J. Cao et al. more recently explored lncRNA expression levels in HCC and para-carcinoma tissue to filter out 19 specific differentially expressed lncRNAs, demonstrating for this 19-lncRNA signature a good diagnostic accuracy in diagnosis and prognosis prediction in patients with HCC (AUC > 0.70) [113].

Altogether, these encouraging findings suggest that ncRNA expression signatures could be considered a potential tool for the diagnosis and prognosis determination of HCC.

### 3.4. Etiology-Specific Dysregulated Circular RNAs in Hepatocellular Carcinoma Progression

In addition to the above-presented evidence composing a clinical–pathogenetic heterogeneous background, identifying an association between definite ncRNAs and specific etiological HCC agents, in the optic of HCC-patient-tailored management, constitutes a current research challenge and would certainly represent a medical breakthrough.

In the context of viral hepatitis, the detailed molecular mechanisms by which HBV contributes to the development of HCC remain largely unknown, and specific aberrantly expressed circRNAs might be involved in the pathogenesis of HBV-associated liver cancer by regulating several tumor-related molecules or pathways [114]. In this regard, a total of 13,124 circRNAs were recently found dysregulated in HBV-related HCC, and notably, the circRNA–miRNA interaction network analysis revealed that 6020 circRNAs were predicted to target 1654 miRNAs [114].

In particular, the downregulation of *circRNA_10156* has been demonstrated to suppress liver cancer cell proliferation, suggesting this molecule is a pro-tumorigenic element [114]. In the same study, the authors clarified the pathogenetic role of this ncRNA and the relatively targeted miRNA, highlighting that circRNA_10156 may act as a molecular sponge of the oncomiR miR-149-3p, which serves a crucial role in tumor development [114]. Consistently, they demonstrated that the depletion of circRNA_10156 upregulated miR-149-3p, reduced Akt1 serine/threonine-protein kinase expression, and suppressed liver cancer cell proliferation [114].

In the last few years, in parallel to viral hepatitis scenarios, given also the epidemiological data, NAFLD-related HCC represents, likewise, a research field where the role of circRNAs has been intensively investigated [115]. In this regard, *circRNA CDR1-AS* has been shown to be upregulated in HCC tissues with simultaneous downregulation of miR-7. This circRNA has been proposed to act as a molecular sponge of TS miR-7; consistently, CDR1-AS-determined decreased miR-7 expression promotes HCC progression facilitating the invasion/proliferation mechanisms of cancer cells [115,116,117].

In the same vein, *circRNA_0067934* can promote progression events associated with HCC worsening as the invasion and metastasis-related mechanisms via the β-catenin/Wnt signaling pathway in the NAFLD setting. In support of this, the enhanced expression of this molecule has been found in HCC tissues compared with peritumoral healthy specimens and its inhibition suppresses the invasion and metastasis of liver cancer [115,118].

In contrast with circRNA_CDR1-AS and circRNA_0067934, *circRNA MTO1*, by acting as the oncomiR miR-9 sponge, has been associated with anti-cancer advancement mechanisms and its upregulation was proposed to negatively regulate HCC progression [115,119].

All these findings demonstrate how a better understanding of the molecular etiological-specific mechanisms of HCC carcinogenesis would contribute to the development of more effective molecular targeted interventions for the primary prevention and treatment of HCC.

Table 1 summarizes lncRNA dysregulated pathways influencing etiologically specific HCC progression mechanisms.

## 4. Non-Coding RNAs and Novel HCC Pathogenetic Frontiers

lncRNAs, along with their smaller counterpart miRNAs, were long considered useless components of DNA sequences, defined indeed as “junk DNA”. Progressively, their potential and function in both physiological and pathological conditions have been displayed by numerous studies explaining their mechanisms of action, including their influence on HCC genesis and progression [83].

In parallel with the classic intrinsic hepatocarcinogenic mechanisms, in recent decades the role of innate and acquired immunity (systemic and local), oxidative stress (systemic and local), as well as the impact of composition and functioning of the gut microbiota on the genesis and progression of HCC, have emerged as important deus ex machina, acting on the complex pathogenetic scenario of liver cancer.

In this regard, a plethora of papers focusing separately on the role of ncRNAs, as well as on the aforementioned new pathogenetic frontiers of HCC, is currently available in the literature.

The novelty of this review consists of exploring the pathogenetic relationship of ncRNAs with immune dysregulation, alteration of the oxidative homeostatic balance and gut dysbiosis, and contextualizing the currently clarified functions of individual non-coding transcripts within this complex network.

### 4.1. Role of Non-Coding RNAs in the HCC Microenvironment: A Matter of Immunity?

Over the last decade, convincing evidence has highlighted the critical role of the tumor microenvironment (TME) as an important determinant in the development of many malignant cancers, including HCC [130,131,132,133]. The liver is a central immunomodulator ensuring a homeostatic balance between immunoreactivity, which confers protection, and immunotolerance, which turns off the immune response [134]. A newer hallmark characterizing the HCC pathogenesis is the deregulation of this tightly controlled immunological network involving the resident and systemic innate immune cells and adaptive immune cells [135]. Specifically, the immune response in the liver is modulated by its continuous exposure to toxic molecules and gut-derived microorganisms and requires a degree of immune tolerance to protect normal tissue from damage. Indeed, there is a strict balance in the production of proinflammatory cytokines (IL-6, TNF-α, IL-2, IL-7, IL-12, IL-15, IFN-γ) and anti-inflammatory cytokines (IL-10/IL-13/TGF-β) to coordinate resident and periphery leukocytes such as T cells and B cells, macrophages [Kupffer cells (KCs)/monocytes], natural killer (NK) cells, natural killer T (NKT) cells, and HSCs.

Many studies focused on the role of several ncRNAs (lncRNAs, miRNAs, and circRNAs) and the competing endogenous RNAs (ceRNAs) forming an intricate regulatory network in the immune cells. This current research pointed out their involvement in the regulation of important mechanisms such as differentiation, activation, and effector functions of both adaptive and innate immune cells. Overall, the scientific assumptions concerning this research field provide a fascinating context to elucidate immune-related molecular mechanisms in HCC with translational applications.

Recently, Zhan et al. identified a novel dysregulated lncRNA-miRNA-mRNA network associated with HCC and its early relapse [136]. LncRNA *SNHG3* fostered the expression level of Programmed cell death protein 1 (PD-1) by regulating anti-silence Function 1B (ASF1B) via sponging of miR-214-3p in HCC, activating tumor immune tolerance and escape. PD-1 is associated with the inhibition of the immune response of HCC patients by negatively regulating the activation and function of T cells, thereby contributing to tumor aggressiveness and postoperative recurrence. Alternatively, ASF1B is a chaperone protein of histone H3-H4, playing an important role in DNA replication, damage repair, and transcriptional regulation [137]. Interestingly, prognostic analysis showed that HCC patients with high ASF1B expression levels had worse disease-free survival (DFS) and higher HR-relapse in the decreased B cells, decreased CD8^+^ T cells, decreased neutrophils, and enriched CD4^+^ T cells subgroups. The latter has been reported to inhibit the proliferation of effector T cells and to boost the evasion of HCC cells from the anti-tumor immune response [138]. Therefore, given the role of ASF1B, it is conceivable to imagine both a new immunotherapy-based strategy and RNA-interference-based strategy inhibiting lncRNA SNHG3 in HCC.

In addition, Huang and coworkers found a novel circRNA (*circMET*, *hsa_circ_0082002*) upregulated in HCC tissues compared to paratumor tissues, which drive immunosuppression and anti-PD-1 therapy resistance inducing a poor prognosis [139]. In particular, data obtained from Chip-seq and luciferase reporter assays showed that circMET overexpression induced an immunosuppressive tumor microenvironment via the miR-30-5p/Snail/DPP4/CXCL10 axis. Hence, circMET overexpression induces the sponging of miR-30-5p resulting in an abnormal post-transcriptional regulation of Snail which acts as a transcription factor of the dipeptidyl peptidase-4 (DPP4). This enzyme is involved in local immunosuppression by negatively regulating lymphocyte trafficking via cleavage of the chemokine CXCL10 [140]. Furthermore, treatment with the DPP4 inhibitor sitagliptin significantly improved the antitumor effects in immunocompetent mice bearing tumors with high levels of circMET and Snail. From a clinical point of view, sitagliptin was shown to enhance CD8^+^ T cell infiltration in HCC patients.

Another circRNA found to be involved in immune evasion in HCC progression was *circGSE1* [141]. CircGSE1 was highly expressed in HCC serum-derived exosomes and through in silico analysis, luciferase reporter assays, and pulldown experiments confirmed its sponging function towards miR-324-5p. Investigating the biological significance of this circRNA-miRNA interaction, the authors showed that circGSE1 leads to immune escape of HCC through regulatory T cells (Tregs) enlargement via regulating the miR-324-5p/TGFβR1/Smad3 axis. The circGSE1-mediated downregulation of miR-324-5p gave rise to an increased expression of TGFB1 which, acting as a TGFβ1 receptor, heightens the signal transduction, triggering several immune pathways in T cells. Alongside TGFB1 upregulation, an increased expression in Smad3 was shown, which, by acting as a TGFβ1 substrate, takes part in the transcriptional regulation of downstream genes [142]. In particular, Tone et al. demonstrated Smad3-induced FOXP3 overexpression through the binding to its enhancer, thereby promoting Treg expansion [143]. Moreover, these mechanisms were shown to be involved in the progression stage of HCC by performing different transwell and wound-healing in vitro assays, but further confirmed through in vivo experiments where circGSE1 facilitated HCC metastasis. Taken together, these data indicate the potential of circGSE1 as a new target for HCC immunotherapy. Whilst all the data discussed above are oriented towards the elucidation of molecular mechanisms involved in the immune escape on an adaptive basis, several studies underlined the importance of the dysregulation of key mechanisms in cells of innate immunity, such as macrophages, which represent the “power switch” of the immune response as a first defense barrier. Indeed, macrophages undergo polarization into the M1-like phenotype, which plays important roles in the inflammatory response and antitumor immunity, or the M2-like phenotype, which is crucially involved in tissue repair and tumour progression [144].

Zhao and coworkers in well-conducted and thrilling research exhibited the intertwining of different epigenetic mechanisms inducing the genesis of tumor-associated macrophages (TAMs) displaying the M2-like phenotype responsible for the establishment of an immunosuppressive microenvironment involved in the HCC progression [145]. Notably, the epigenetic silencing of the *miR-144/451a* cluster contributes to HCC advancement via paracrine Hepatocyte Growth Factor (HGF)/Macrophage Migration Inhibitory Factor (MIF)-mediated TAM remodeling [146]. This miRNA cluster correlates with a better prognosis and enhancement in clinical features, but mechanistically, it is transcriptionally repressed by EZH2-catalyzed histone H3K27 methylation of the promoter and by the DNA methylation of the distal enhancer in HCC cells [146].

Basically, these mechanisms have a strong impact on tumor microenvironment remodeling with an increase in HGF and MIF expression both involved in the promotion of M2 polarization of macrophages, thereby participating in the anti-inflammatory response in various tissue and facilitating tumor progression. However, mir-144/miR-451a overexpression was shown to impair the M2 phenotype and, conversely, to be determinant in the stimulation of M1 repolarization of TAMs by targeting the secretion of HGF and MIF from HCC cells resulting in restored phagocytosis and an enhanced capability to activate cytotoxic T lymphocytes [146]. Therefore, considering the chromatin conformation remodeling and non-coding-RNA-mediated mechanisms in HCC patients could provide new insights for a better understanding of pathogenesis and diagnostic strategies.

In the same vein, Han et al. identified another critical player involved in macrophage-based immune escape and HCC progression [147].

*LncRNA TUC339* is highly expressed in HCC-derived exosomes and induces both tumor growth and metastasis along with macrophage regulation of M1/M2 polarization dampening the anti-tumor immune response in vitro [148]. Data obtained from microarray studies revealed the downregulation of the Toll-like receptor (TLR) signaling and FcR-mediated phagocytosis pathways induced by TUC339. Interestingly, TUC339 knockdown increases macrophage phagocytic activity and changes in cytokines and chemokines pathways [149], although further research is needed to clarify the precise mechanisms.

Additionally, recent rousing research conducted by Liu et al. pointed out the pivotal role of Akt activation in establishing an M2 phenotype in KCs in vivo impairing hepatic enrichment of CD8^+^ T cells [150]. Considering the extreme association of the AKT/mTOR and RAS/MAPK cascades with biological aggressiveness and poor prognosis of HCC, the authors reproduced in vivo this phenotype by hydrodynamically transfecting activated forms of AKT (myr-AKT) and NRas (NRas-V12) oncogenes (AKT/Ras) into the mouse liver. Akt signaling in Kc was shown to prompt the dysregulation of different miRNAs [150]. Among these, *miR-206* was found to drive M1 polarization of KCs and to be essential for the activation of CCL2/CCR2 signaling to facilitate hepatic recruitment of cytotoxic T lymphocytes [150]. Based on the capability of miR-206 to prevent HCC by enhancing immune surveillance, it represents a potentially novel immunotherapeutic agent against liver cancer.

Altogether, exploring the multiple roles of ncRNAs in immune-related mechanisms involved in HCC emphasizes the value of a deeper investigation of the molecular pathways involved in innate and adaptive immune escaping. The advances in this research field could allow in the near future a potential tailored immunotherapy that would revolutionize the clinical therapeutic approach to HCC.

Figure 3 represents the most relevant nc-RNAs regulating dysregulated immune pathways involved in HCC onset and progression.

Overexpression of lncRNA SNHG3 induces miR-214-3p sponging and downstream anti-silence Function 1B (ASF1B) and PD1 upregulation. Accordingly, decreased CD8^+^ T cell, decreased neutrophil, and enriched CD4^+^ T cell subgroups contribute to tumor aggressiveness. circMET overexpression induced an immunosuppressive tumor microenvironment through the sponging of miR-30-5p and the upregulation of Snail, which induced dipeptidyl-peptidase-4 (DPP4) expression. DPP4 downregulates the chemokine CXCL10, reducing lymphocyte trafficking. circGSE1-mediated downregulation of miR-324-5p increases expression of TGFB1, which through Smad3 and FOXP3, trigger Treg expansion. The epigenetic silencing of the miR-144/451a cluster induced by EZH2-catalyzed histone H3K27 methylation increases HGF and MIF expression promoting M2 polarization and the genesis of tumor-associated macrophages (TAMs). Akt-mediated downregulation of miR-206 impairs CCL2/CCR2 signaling, facilitating hepatic recruitment of cytotoxic T lymphocytes.

### 4.2. Non-Coding RNAs and Oxidative Stress: “Ferroptosis” Influences Immune-Mediated Progression

Over recent years, ferroptosis, an interesting type of apoptosis, has progressively emerged. Ferroptosis is a recently highlighted form of reactive oxygen species (ROS)-mediated programmed cell death, which is dependent on iron metabolism and lipid peroxidation [151]. This process induces the accumulation of lipid peroxidation products, which, in turn, lead to programmed cell death.

The importance of ferroptosis has been demonstrated in the regulation of metabolism and redox biology, affecting the pathogenesis and treatment of cancers, including HCC. In addition, Liu et al. described a ferroptosis- and immune-related signature and found that this prognostic signature could be used to screen HCC patients for immunotherapies and targeted therapies [152].

Intriguingly, emerging evidence has shown the potential of lncRNAs in regulating ferroptotic cell death for cancer biology. Hence, deepening our understanding of the role of these ncRNAs seems to be a promising tool in the understanding of this particular type of apoptosis in HCC. An investigation carried out by Zhu and colleagues in 2021 showed how the molecular profile of lncRNAs in plasma samples of HCC patients is altered and mainly enriched in biological activities like tumorigenesis, cell metastasis, and immune response [153]. Furthermore, Xu and other researchers identified 3714 genes (3433 upregulated and 281 downregulated) that were differentially expressed in the TCGA-HCC dataset [154]. Among these deregulated genes, a total of 24 DE-lncRNAs (LINC00942, LINC01224, LINC01231, LINC01508, etc.) were determined as the ferroptosis-related lncRNAs and tightly associated with ferroptosis-associated genes [154]. Kaplan–Meier survival analysis was further used to evaluate the importance of lncRNA expression on the prognosis of patients and high levels of these candidate DE-lncRNAs were all correlated with poor prognosis in patients with HCC [154].

Moreover, a different analysis demonstrated how ferroptosis is becoming a crucial factor in the prognosis of patients with HCC and other malignancies. The induction of ferroptosis is closely correlated to anti-tumor immunity, not only engaging in tumor cell destruction through Immune checkpoint inhibitors (ICI)-activated T cells but also directly altering the function of diverse immune cells, implying the prospect of cancer synergistic therapy [155,156,157,158].

At the same time, new findings show that several lncRNAs can play an important role in regulating the occurrence and development of diseases by promoting ferroptosis [156,157,158,159]. Yang and others elucidate a lncRNA prognostic signature model of 17-ferroptosis-related lncRNA prognostic which could reliably predict the prognosis of HCC patients. They further explored the link between ferroptosis and ICIs to demonstrate how in high-risk HCC patients, ferroptosis-related lncRNAs, together with the canonical ICIs may promote malignant cell ferroptosis, thereby improving overall prognosis [159]. This research has pointed out a fascinating liaison between traditional and emerging new fields of therapy which need to be further deepened to improve anti-tumor activity synergistically, even in ICI-resistant types, giving a pivotal role to the combination of ICIs and ferroptosis inducers as a possible novel therapy options for HCC patients in the future.

### 4.3. Gut Microbiota and Immunity: A Pathogenetic Bridge on HCC Landscape

The human gastrointestinal tract harbors more than 1000 different bacterial species that reside within and colonize it (“*autochthonous bacteria*”) or pass transiently through it (“*allochthonous bacteria*”) but most of them colonize the human colon [160]. Five bacteria phyla can be considered the dominant microflora: *Firmicutes*, *Bacteroides*, *Actinobacteria*, *Proteobacteria*, and *Verrucomicrobia* [160]. In the condition of eubiosis, the autochthonous intestinal bacteria are involved in the catabolism of several elements derived from the diet or endogenous secretions [160]. Thanks to the by-products, they can modulate the expression of host genes participating in several pathological functions and also interfere with the immune system [160]. In addition, the intestinal microbiota is involved in the inflammation mechanism, redox stress damage, motility, angiogenesis, proliferation, differentiation, fat storage regulation, carcinogenesis, cancer response to chemotherapy, and even cognitive function [161,162].

Several studies have shown the emerging role of dysbiosis in the pathogenesis of several diseases, including liver disease and, in particular, the development of HCC [163,164]. Indeed, evidence suggests that the gut microbiota can increase lipopolysaccharide (LPS) levels and can establish a consequential pro-inflammatory microenvironment in the liver. Concerning the contribution of the gastrointestinal tract to preserving homeostasis by maintaining an intact barrier against LPS and intestinal bacteria, Wan et al. showed how leaky gut, endotoxemia, TLR, dysbiosis, and immunomodulation promote the development of HCC [165]. In patients with chronic liver disease or LC, detoxification, degradation, and clearance of LPS and other bacterial products are compromised [165].

In general, patients with HCC have altered microbiota, specifically in high levels of *Escherichia coli* and other Gram-negative bacteria that are associated with increased LPS serum levels [166]. Conversely, the levels of *Lactobacillus* spp., *Bifidobacterium* spp. and *Enterococcus* spp. are reduced in the gut microbiota of HCC patients. An interesting study by Lu H. et al. showed how the microbial metabolism, iron transport, and energy-producing system are significantly different between the microbiota of HCC patients and healthy controls [167]. Moreover, the TLR-9 signaling pathway induces the production of IL-1β by KCs, leading to steatosis, inflammation, and fibrosis [168,169].

In 80% of cases, HCCs develop in a microenvironment characterized by chronic injury, inflammation, and fibrosis; however, there is still much to be studied about mediators that are involved in a high risk of developing HCC. Several lines of evidence show how the microbiota and TLR pattern link inflammation and carcinogenesis in the chronically injured liver [168]. In this context, a recent study indicates a relevant role of TRL signaling in the progression of NAFLD and NASH, revealing NASH patients’ higher prevalence of Small intestinal bacterial overgrowth (SIBO), which is associated with enhanced expression of TLR-4 in the liver [170]. Conversely, probiotic treatment increased anti-inflammatory cytokine secretion in a TLR-2-dependent manner, while *Clostridium butyricum* induced IL-10 production from intestinal macrophages in acute experimental colitis through TLR-2; this suggests that TLR-2 has a dual function: TLR-2 ligands from probiotic bacteria are anti-inflammatory, whereas TLR-2 ligands from the bacteria that are present during obesity induce inflammation [170]. Therefore, the gut microbiota may be involved in the progression of NAFLD and NASH by regulating TLRs [171,172].

Similarly, Dapito D. H. and his research groups, in a study with a mouse model demonstrated that TLR-4-dependent HCC promotion in the early phases of hepatocarcinogenesis is predominantly mediated by TLR-4-dependent secretion of growth factors such as epiregulin by HSCs [173]. During the later stages of hepatocarcinogenesis, they instead detected decreased induction of nuclear factor kappa-light-chain-enhancer of activated B cells (NF-kB)-regulated genes and an increased rate of apoptosis in all groups that were protected from HCC development [173]. These findings demonstrate how the intestinal microbiota drive HCC promotion but not initiation.

### 4.4. NC-RNAs, Gut Microbiota, and Immunity in HCC

There is still much that is unknown about the alteration caused by microbiota to the tumor microenvironment by its effect on circulating ncRNAs such as circ-RNAs/miRNAs that contribute to cancer metastasis. Fascinating research by Zhu et al. indicates a regulatory function of circulating miRNA networks in a gut-microbiota-dependent manner [174]. Specifically, they examined the role of antibiotic (ABX) (broad-spectrum antibiotics)-mediated depletion of the gut microbiota, finding that ABX increased cancer metastasis.

Furthermore, they analyzed the gut microbiota in deep sequencing combined with animal models and fecal transplantation, identifying a critical role of gut microbiota in the regulation of cancer metastasis through the IL-11/circRNA/miRNA/SOX9 axis [174].

Similarly, lncRNAs are emerging as a factor regulating microbiota composition and host–microbiota interaction. As reported by a recent study, lncRNAs can act as molecular signatures in gut tissues and their spatiotemporal expression patterns could be of service as biomarkers to discriminate a healthy gut from a dysbiotic gut [175]. Wen and collaborators found 75 lncRNAs and 49 lncRNAs in probiotic- and pathogen-mediated carcinogenesis thanks to RNA sequencing analysis [176].

The possibility that lncRNAs are linked to host–pathogen interaction is a promising target for the development of diagnostic and prognostic biomarkers. Indeed, some obligate intracellular pathogens and parasites infect via exosomes that are internalized by cells to form endosomes [177].

Regarding the new vision of metabolic liver disease, which is recognized as a risk factor for HCC, several studies showed that different highly conserved miRNAs play critical roles in controlling metabolic homeostasis and their dysregulation contributes to the development of obesity and insulin resistance [178,179]. In this context, Virtue and his collaborators demonstrated that in white adipocytes, the expression of the miR-181 family is regulated by the gut microbiota during homeostasis to regulate key pathways involved in the control of adiposity, insulin sensitivity, and white adipose tissue (WAT) inflammation in mice [180]. Therefore, considering what has been revealed, it is necessary to increase studies on the link among microbiota, immunity, and ncRNAs to identify unknown epigenetic biomarkers and pathways useful to determine disease progression as well as use them as therapeutic targets. Concurrently, the gut microbiota could be a new target for investigation to increase our knowledge about its role in immunity modulation and find new therapeutic strategies.

## 5. Potential ncRNA Applications in HCC Treatment: A Close-to-Exploding Time Bomb or an Under-Built Sand Castle?

In the modern panorama of Translational Medicine, one of the most promising practical applications of ncRNAs appears to be therapeutics. Theoretically, in particular for sncRNAs, as it is possible to chemically generate siRNAs and miRNAs targeted against any cellular RNA, these molecules represent potentially usable “weapons” to downregulate the expression of specific gene-causing diseases [181]. At the same time, by similarly acting, specific related lncRNAs and sncRNAs have been demonstrated to be actively involved in cancer-related chemoresistance processes, representing potential tools to optimize therapeutic approaches even in HCC management [72]. Based on this, in the last few decades, various encouraging results have emerged from the dramatic amount of research exploring potential ncRNA HCC-therapeutic targets [182,183].

However, despite the great therapeutic potential, the use of siRNAs/miRNAs as anti-HCC agents has for a long time not achieved many breakthroughs due to the initial lack of efficient delivery systems. In effect, these molecules have to face a selective obstacle course before reaching their specific intracellular target. In particular, when introduced as free molecules, these sncRNAs are rapidly degraded by nucleases, cleared by the kidneys, and cannot efficiently cross cellular membranes [181]. Moreover, after cellular uptake, siRNAs/miRNAs can be sequestered into endosomes losing the possibility of reaching the target RNA into the cytoplasm/nucleus [181]. Last but not least, it is extremely important to avoid the off-targeting effect, by specifically driving siRNAs/miRNAs in the diseased tissue rather than in healthy tissue [181].

All these features justify the adoption of delivery systems based on the use of viral vectors or synthetic vectors which, progressively developed over the last few years, have made it possible to overcome the abovementioned obstacles. In general, despite the fact that practical difficulties in siRNA/miRNA delivery for HCC persist, the increasing scientific attention in this field represents the fuel to simultaneously design novel technical solutions.

The next paragraph, by reviewing the most innovative and relative studies on the topic, aims to present an overview of the main applications of ncRNAs in HCC treatment.

### 5.1. SncRNAs’ Therapeutic Applications for HCC

*SiRNAs* are short synthesized RNAs deriving from extracellular sources able to silence specifically gene expression via target RNA degradation [181]. In effect, the targeting of molecules more relevant for HCC cell survival than for normal hepatocytes and genes over-expressed in HCC compared to normal liver have long been recognized as the two guiding strategies in the novel therapeutic frontiers of HCC.

Regarding the first approach, in the last decade, integrins have been identified as fundamental proteins in the HCC microenvironment, representing extracellular matrix receptors with important and different involvement in the regulation of cell motility, survival, and proliferation, functioning even in non-neoplastic cells [184,185]. Therefore, the silencing of specific RNA-codifying integrins mediated by opportunely vectorized siRNA has represented one of the most investigated therapeutic applications of these sncRNAs in the treatment of HCC [181].

For this purpose, Bogorad et al. have investigated multiple chemically modified (to improve the stability and reduce the off-target potential) anti-integrin siRNAs, delivered by nanoparticles made by ionizable lipid or cationic lipid (used to form spontaneous complexes with siRNA) in HCC-affected rat models [186]. However, although in these experimental conditions, the reduction in cell proliferation rhythm and increase in tumor cell death suppressed HCC progression, these were accompanied by long-term adverse effects observed after integrin silencing, suggesting the need to focus efforts on the second abovementioned and more selective strategy [186].

Concerning the second approach, Chen et al. demonstrated that the downregulation of Epithelial cell transforming sequence 2 (ECT2), a protein regulating cycle cell progression resulted in overexpressed in HCC cells, obtained through ECT2 targeting by using siRNAs delivered via a viral vector, significantly reduced tumor growth in a subcutaneous xenograft mouse model of HCC, although a prolonged survival was not observed [187]. Once again, in a subcutaneous xenograft mouse model, the administration by direct tumor injection of the anti-IFN-stimulated genes 15 (ISG15) siRNA, considering ISG15 overexpression in HCC and the reported relative association with abnormal cell signaling and malignant transformation, suppressed tumor growth, also determining in this case, a prolonged animal survival of about 15% [188].

Moreover, in vitro and in a murine model of HCC, the aberrant proliferation of HCC cells determined by Myc overexpression was efficiently downmodulated by siRNA selectively silencing CDK9, suggesting CDK9 inhibition as a valid strategy for Myc-overexpressing in hepatocancerogenesis [189].

Altogether, these encouraging results opened the doors to advanced experimental steps in the research field of siRNAs in HCC, and many siRNA-based drugs have even been progressively considered in clinical trials. For example, in a recent Phase II clinical trial, Zuckerman et Davis investigated the anticancer drug DCR-Myc, a lipid-nanoparticle–formulated siRNA targeting Myc, in patients with HCC [190]. In another Phase II clinical trial, Yamada et al. studied TKM-PLK1, a lipid nanoparticle encapsulating siRNA against the polo-like kinase-1 (PLK1) gene product, revealing the overexpression of the relative encoding gene in liver cancer and its involvement in metastases formation [191]. Thus, the silencing of PLK1 activity in proliferating hepatocytes appears to rapidly induce cell-cycle arrest and apoptosis [191].

Regarding *miRNAs* as potential therapeutic targets in HCC, as reported in previous paragraphs, these ncRNAs play a not-unidirectional role in cancerogenesis, considering the context of oncomiRs and TS-miRs simultaneously functioning in liver cancer. In light of this, if we consider that miRNA expression levels can be either inhibited or enhanced to achieve tumor suppression, the therapeutic potentialities of the modulation of these molecules appear even more interesting than siRNAs, whose unique philosophy of action is silencing and inhibiting specific molecules [181,182].

Although every dysregulated miRNA would theoretically represent a treatment target, we herein now report the most investigated experimental attempts that concretely evaluated the applications of these molecules. The delivery of miRNA mimics, replacing and enhancing the function of the original molecule, represents the more consolidated experimental HCC-ncRNAs’ therapeutic frontier [181]. In this context, the administration of *miR-122* mimic demonstrated favorable effects on HCC progression by reducing cell proliferation and angiogenesis [192]. The consequences of miR-122 downregulation in HCC contexts have been widely investigated. Therefore, it is reasonable to expect several encouraging results from its transfection. Consistent with this, in an in vitro study, the LNP-DP1 (a cationic lipid nanoparticle formulation)-mediated transfection of *miR-122* mimic, by regulating various crucial target oncogenes in HCC cells, suppressed cell proliferation, neo-angiogenesis, and thus, cancer progression [193]. In addition, another interesting potential application using *miR-122*, considering the enhanced capacity of liver cells to internalize short molecules during this locoregional procedure, is represented by the injections of this molecule in the course of trans-arterial chemo-embolization (TACE) [194].

Moreover, another interesting mimic miRNA is the *miR-34a* mimic “MRX34”, which has been recently evaluated in a clinical trial (NCT01829971) assessing the effects on different advanced solid tumors, although it should be noted that because phase I was complicated by severe adverse effects, phase II of the study provided an adjustment in dose administration for HCC [195,196]. Furthermore, another emerging perspective is the adoption of nanomaterials as miRNA therapeutic agents aiming to deliver therapeutic agents that can stimulate endogenous miRNAs [197]. For instance, the utilization of gold nanoparticles (AuNPs) that transfer *miR-326* mimic promoted its overexpression and the suppression of the PDK1/AKT/c-Myc axis, blocking invasion–migration processes and increasing apoptosis [198].

Finally, as well as by synthetic vectors, the delivery of miRNAs can also be achieved by using adeno-associated virus (AAV) vectors. Regarding this, in an HCC animal model, the systemic administration of TS-miR-26a delivered by AVV vectors significantly suppressed tumor progression by promoting apoptosis and blocking cell proliferation (targeting cyclin D2 and E2) [55]. Considering the absence of reported related toxicity, these encouraging results appear very relevant and lay the foundations for future human applications.

### 5.2. ncRNAs: Small Molecules Greatly Contribute to the HCC Chemotherapy Resistance

Despite surveillance programs worldwide currently allowing early HCC identification in 40–50% of cirrhotic individuals, in a non-negligible number of patients, HCC diagnosis is gained at a clinical-course time-point when only potentially curative treatments are applicable. In addition, almost half of all HCC patients in advanced tumor stages, due to the progression/recurrence of disease, ultimately require systemic therapies [199]. Advances in immunotherapy for HCC have also added hope for patients, but their efficacy remains limited [200]. Therefore, in all these cases, systemic chemotherapy (SC), including in primis, the approved tyrosine-kinase-inhibitor drugs (TKIs) (Sorafenib and Lenvatinib overall) often represent the final therapeutic bastion for patients with advanced HCC (BCLC-C) [199]. Unfortunately, as the efficacy of SC is frequently invalidated by drug resistance development contributing to treatment failure, significant long-term benefits, both in terms of Performance Free Survival (PFS) and Overall Survival (OS), are recurrently not appreciable [199]. In this sense, the improvement of treatment outcomes for SC-treated HCC patients appears strictly bound to the comprehension of molecular mechanisms contributing to drug resistance acquisition whose clarifications would represent a cornerstone in the development of more efficacious therapeutic strategies.

In the last few years, a growing number of findings suggesting an active involvement of ncRNAs in HCC-related chemoresistance mechanisms have been reported [72].

*Sorafenib* is a multikinase TKI targeting several oncogenetic factors promoting cancer cell proliferation and angiogenesis, including v-RAF murine sarcoma viral oncogene homolog B (BRAF), VEGFR-2/3, and platelet-derived growth factor receptor (PDGFR) [201]. HCC resistance to Sorafenib is supported by robust evidence and the implications of several ncRNAs in chemoresistance mechanisms have been widely suggested [72]. In an In vitro study, *miR-181* was shown to induce Sorafenib resistance via suppressing Ras association domain family 1 (RASSF1), a MAPK signaling factor [202]. Moreover, *miR-494*, an oncoMiR that promotes cell proliferation, migration, and invasion, was also shown to increase Sorafenib resistance in HCC by targeting Phosphatase and tensin homolog (PTEN) [203]. In addition, PTEN was revealed as the functional target of the *miR-216a/217* axis whose overexpression acted as a positive feedback regulator for the Transforming Growth Factor (TGF)-β pathway and the pathway involved in the activation of phosphoinositide 3-kinase/protein kinase K (PI3K/Akt) signaling in HCC cells [204].

In support of this, the activation of the TGF-β- and PI3K/Akt-signaling pathways resulted in acquired resistance to sorafenib; in contrast, blocking the activation of the TGF-β pathway overcame *miR-216a/217*-induced Sorafenib resistance [204].

In light of these findings, the Akt pathway appears crucial in modulating HCC cell sensitivity to Sorafenib, which is consistently supported by the evidence that the lncRNA NEAT1 overexpression modulates Sorafenib resistance via regulation of the *miR-149-5p*/AKT1 axis [205].

*Lenvatinib* is a small TKI molecule inhibiting different receptor targets crucially involved in tumor angiogenesis and proliferation such as VEGFR1-3, fibroblast growth factor receptor (FGFR1-4), PDGFRα, stem cell factor receptor (KIT), and rearranged during transfection (RET) [206].

The REFLECT trial, by demonstrating Lenvatinib’s non-inferiority, but not superiority, to Sorafenib in the improvement of OS, proposed a long-awaited alternative to Sorafenib for first-line targeted therapy of patients with advanced HCC [207]. However, consistent with the other TKIs, resistance to Lenvatinib leads to tumor progression and constitutes a major obstacle to improving the prognosis of HCC patients.

Several ncRNAs have been shown to play a crucial role in the acquisition and maintenance of HCC cell Lenvatinib resistance, in particular by interfering with HCC cell apoptotic mechanisms.

In this context, in an interesting study, Ting Yu et al. revealed the dysregulation in Lenvatinib-resistant-HCC cells of the lnc-RNA *MT1JP* whose overexpression determined the inhibition of the apoptosis signaling pathway [208]. Furthermore, the authors found that the sponging of microRNA-24-3p by *MT1JP* promotes the releasing of Bcl-2 like 2 (BCL2L2), an anti-apoptotic protein, configuring a novel molecular loop between *MT1JP* and apoptosis signaling [208].

In the same vein, *miR-128-3p* was shown to mediate Lenvatinib resistance of HCC cells in vitro by downregulating c-Met; in particular, *miR-128-3p* and c-Met were revealed to participate in the mechanisms fueling Lenvatinib resistance impacting on the PI3K/AKT apoptotic-regulating pathway [209]. Complementarily, Han et al., demonstrated that *miR-183-5p*, by directly targeting and downregulating the cell surface-associated protein mucin 15 (MUC15), which normally interacts with c-MET leading to the inactivation of the PI3K/AKT signaling pathway, promotes liver tumorigenesis and HCC cell Lenvatinib-resistance [210].

More recently, the overexpression of the lncRNA *HOTAIRM1* was shown to favor Lenvatinib resistance by downregulating *miR-34a* and activating autophagy in HCC cells [211]. In vivo/in vitro findings revealed that the knocking down of *HOTAIRM1* increased the expression of *miR-34a* in Lenvatinib-resistant cell lines, as well as increasing their sensitivity to Lenvatinib, especially when combined with autophagy inhibitors [211].

To close the circle surrounding the centrality of ncRNAs in liver cancer chemoresistance, we now present the other side of the coin by reviewing the potential applications of certain miRNAs and siRNAs that have been demonstrated to suppress some of the crucial mechanisms inducing TKIs-resistance in HCC cells.

Rudalska et al., by first developing a novel siRNA-based approach to screen in vivo genes whose inhibition increases the therapeutic efficacy of Sorafenib, identified the kinase Mapk14 as a key element of resistance to this TKI in mouse liver cancer [212]. Of relevance, in animals, the silencing of this kinase mediated by specific Mapk14-silencing-siRNA sensitized the mice to Sorafenib treatment, determining reduced tumor growth and longer survival in comparison with animals treated with Sorafenib alone [212]. Moreover, the absence of significant toxicity related to this treatment reported in this research may pave the way to its potential application in human HCC treatment.

In the same vein, Zhang et al. explored the consequences of Aurora-A silencing by siRNAs in HCC scenarios in vitro [213]. Aurora A is a member of the Aurora kinases family, a group of threonine kinase playing a crucial role in mitosis and tumorigenesis whose expression levels has been reported to be higher in HCC contexts [213]. Interestingly, firstly in HCC cell lines and, subsequently, in a confirmatory xenograft mouse model, Aurora-A silencing significantly reduced colony formation and promoted apoptosis, simultaneously improving the chemosensitivity to several chemotherapeutic agents via the increase of apoptosis [213].

Finally, more recently, Dong et al. revealed that *miR-124-3p* downregulation is associated with Sorafenib resistance in HCC cell lines involving the stress transcriptional factor FOXO3a pathway, as well as that the administration of the combination “miR-124-3p mimics plus Sorafenib” significantly enhanced the curative efficacy of this TKI in a nude mouse HCC xenograft model [214].

In conclusion, in HCC management, the prediction of response to treatment, in the optic of a personalized approach that allows choosing the most appropriate chemotherapy regimen for each individual and/or avoiding overtreatment attitudes when non-responsiveness to multiple drugs occurs, represents an unmet need. In this scenario, the identification of novel ncRNAs as predictive therapeutic response biomarkers and as molecules to target to suppress chemoresistance mechanisms represents a promising field of research whose practical application in routine clinical practice appears an ever-closer horizon [215,216].

## 6. Conclusions and Future Directions

In the era of Precision Medicine, considering the above-reported ncRNAs’ involvement in post-transcriptional regulation of gene expression processes, these molecules have been widely proposed as candidate biomarkers for HCC diagnosis and prognosis [217]. At the same time, in a translational medicine view based on the concept of “from bench to bedside”, the elucidation of specific molecular mechanisms regulated by ncRNA functioning may also guide the clinician in choosing the most appropriate personalized treatment.

In this review, we have reported the novel pathogenetic frontiers of HCC progression: immune response, gut microbiota composition, and specific oxidative-stress regulated mechanisms of cell survival mutually influence liver cancer worsening.

Of relevance, ncRNAs play a critical role in this complex scenario and the dysregulation in the expression of these molecules is a key driver element in the progression of this neoplasm. As a consequence, the advances in this research field could allow in the near future, a potential tailored immunotherapy that would revolutionize the clinical therapeutic approach of HCC. In parallel, ncRNAs may variously influence therapeutic strategies in daily routine practice because of their crucial involvement in drug resistance processes. In this regard, several studies have highlighted the key roles of ncRNAs in the chemoresistance of HCC, suggesting these molecules to be the potential promising therapeutic targets for overcoming drug resistance in the treatment of HCC [218].

ncRNAs, indeed, appear able to regulate several mechanisms surrounding HCC drug resistance, including the increased expression of drug efflux transporters that recognize and pump out anticancer drugs out of tumor cells, redistribution of intracellular accumulation of agents, inactivation of apoptosis signaling pathways, enhanced DNA damage repair capacity, accelerated drug metabolism, and activation of cancer stem cells (CSCs) [219].

In this sense, looking ahead, despite the even higher obstacle to overcome, remains the difficulty in selecting critical target ncRNAs from among the numerous candidates, the therapeutic scenario for patients with advanced-not-loco/regional-approachable HCC could be represented by the combination of conventional chemotherapy with adequately vectorized ncRNAs, representing a promising alternative to reduce drug resistance.

## Figures and Tables

**Figure 1 cancers-15-05178-f001:**
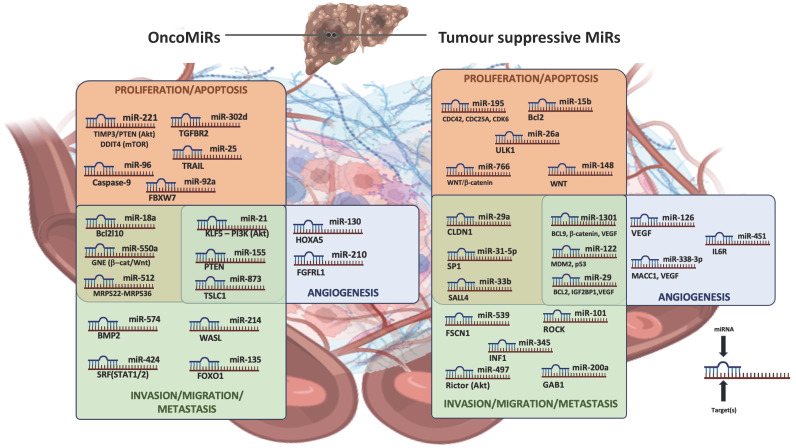
Principal miRNAs with relative targets involved in HCC-progression mechanisms.

**Figure 2 cancers-15-05178-f002:**
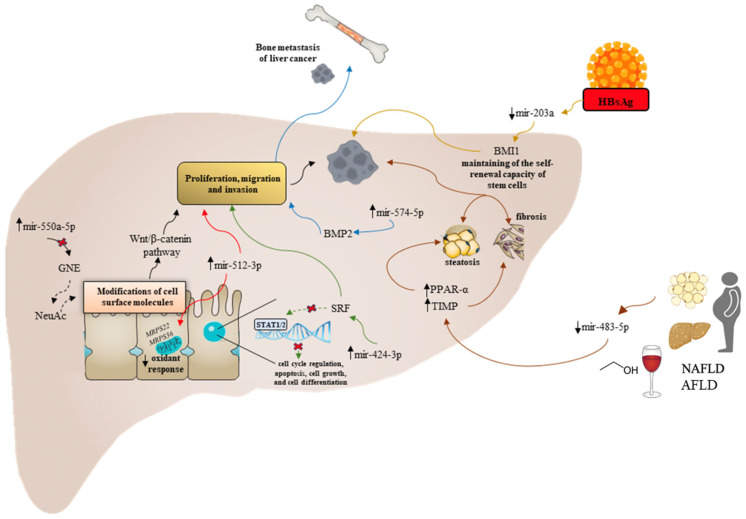
Overview of principal dysregulated miRNAs and related signaling pathways involved in HCC proliferation, migration, and invasion.

**Figure 3 cancers-15-05178-f003:**
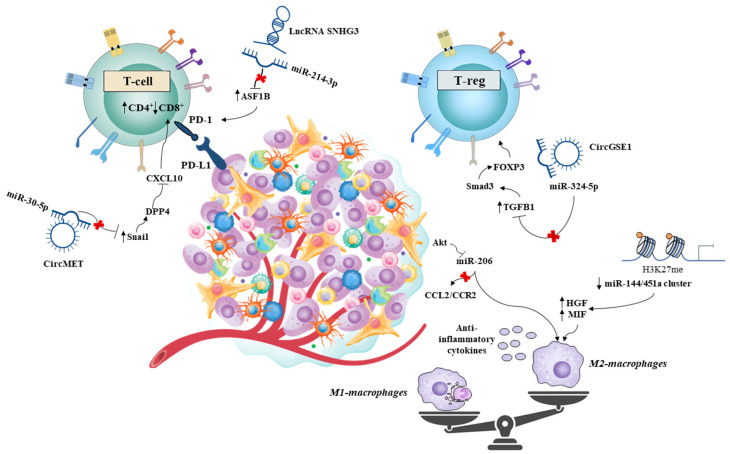
Non-coding RNAs in HCC microenvironment: intricate regulatory networks lead to adaptive and innate immune escape.

**Table 1 cancers-15-05178-t001:** Major long non-coding RNAs associated with hepatocellular carcinoma (HCC) progression in Non-alcoholic Fatty Liver and viral hepatitis with relative mechanisms of action.

		Molecules’ Name	Biogenesis/ Expression Status	Influence on Molecular Pathways Promoting HCC Progression Mechanisms	microRNA(s) Targeted
Non-alcoholic Fatty Liver Disease (NAFLD)—HCC-related context	Linear Long Non-coding RNAs	NEAT1	Overexpression [97,98]	Promotion of NAFLD-related fibrosis worsening and HCC cell proliferation by sponging miR-506 [normally down-regulating acetyl-CoA carboxylase (ACC) and fatty acid synthase (FAS) expression] [97].	miR-506 [98]
		MALAT1	Overexpression [99]	Upregulation of the splicing factor “SRSF1” and activation of the β-catenin/Wnt pathway [99].	Undefined/NA in this context
		CASC2	Downregulation [96]	Regulation of cell proliferation and dysregulation by sponging the oncoMiR miR-155 [96]	miR-155 [96]
	Circular Long-Non-coding RNAs	circRNA CDR1-AS	Overexpression [117]	Promotion of invasion/proliferation mechanisms of liver cancer cells by sponging miR-7 (normally inhibiting spindle checkpoint protein “SPC24”) [117].	miR-7 [116]
		circRNA_0067934	Overexpression [118]	Promotion of invasion and metastasis-related mechanisms by sponging miR-1234 (normally regulating the β-catenin/Wnt signaling pathway) [118].	miR-1234 [118]
		circRNA MTO1	Downregulation [119]	Promotion of HCC progression by sponging the oncomiRNA “miR-9” [119].	miR-9 [119]
Viral-hepatitis—HCC-related context		HBx-LINE1 lncRNA	Integration of HBV DNA in ch.8p11.21 [102]	In HBV context: activation of the Wnt signaling pathway; acting as a decoy to sequester miR-122 [101,102].	miR-122 [101]
		HULC	Overexpression induced by HBx protein [91,103]	In HBV context: activation of sphingosine-kinase-1-mediated angiogenesis, functioning as a molecular decoy of miR-107 (normally repressing the expression of the transcription factor E2F1) [103].	miR-107 [92]
	Linear Long Non-coding RNAs	Linc01419	Overexpression [104]	In HCV context: ○ Promotion of cell proliferation and metastasis by sponging miR-485-5p (normally downregulating “LSM4” expression) [120]. ○ Promotion of cell proliferation and metastasis by enhancing NDRG1 promoter activity [121]. ○ Promotion of cell proliferation by recruiting XRCC5 and regulating its phosphorylation to repair DNA damage [122]. ○ Promotion of proliferation by targeting EZH2-regulated RECK expression [123].	miR-485-5p [120]
		LncAK021443	Overexpression [124]	In HCV context: promotion of cell proliferation, invasion, and metastasis by repressing epithelial-mesenchymal transition (EMT) [104,125]	Unidentified/NA in this context
		Linc01152	Overexpression [126]	In HBV context: increases cell proliferation by activating the STAT3 pathway [126].	Unidentified
Viral-hepatitis—HCC-related context	Circular Long-Non-coding RNAs	circRNA_10156	Overexpression [114]	In HBV context: promotes cell proliferation by sponging miR-149-3p (usually down-regulating the AKT1/mTOR pathway) [114].	miR-149-3p [114]
		circ-RNF13 [circ_0067717]	Downregulation [127]	In HBV context: promotes HCC progression by sponging miR-424-5p (usually regulating TGFβ-induced factor homeobox 2 (TGIF2) [127].	miR-424-5p [127]
		circ_0027089	Overexpression [128]	In HBV contexts: acts as an oncogene and promotes the development of HBV-related HCC by regulating nucleus accumbens associated protein 1 (NACC1) via competitively targeting miR-136-5p [129].	miR-136-5p [129]

HCC: Hepatocellular carcinoma; HBV: Hepatitis B virus; HCV: Hepatitis C virus; NA: not-applicable; NDRG1: N-Myc Downstream Regulated 1; XRCC5: X-ray Repair Cross Complementing 5; EZH2: Enhancer of Zeste 2 Polycomb Repressive Complex 2 Subunit.

## Data Availability

The data can be shared upon request.

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
