# Peer review of "Role of Non-Coding RNAs in Hepatocellular Carcinoma Progression: From Classic to Novel Clinicopathogenetic Implications"

_cancers, 2023, doi:10.3390/cancers15215178_

Round 1

Reviewer 1 Report (Previous Reviewer 3)

Comments and Suggestions for Authors

I thank the editors for the opportunity to review this manuscript again, and the authors for providing a revision according to my own and the other reviewer's suggestions. The authors have fully responded to my feedback and recommendations. Accordingly, the manuscript can be accepted in the present form.

Reviewer 2 Report (Previous Reviewer 1)

Comments and Suggestions for Authors

The authors improved initial version of the manuscript and have added some useful information. 

Comments on the Quality of English Language

English is appropriate with some minor typos and etc

This manuscript is a resubmission of an earlier submission. The following is a list of the peer review reports and author responses from that submission.

Round 1

Reviewer 1 Report

Comments and Suggestions for Authors

The review is devoted to the very wide topic - the role of non-coding RNAs in HCC. To date there are enormous number of publications in the field, both because of hundreds and even thousands of HCC-associated ncRNA   as well as different aspects of HCC.

The authors slightly focused on NAFLD however, at least it was claimed in the summary and the introduction, however the review didn’t save this focus and have a lot of rather general information.

Major points:

  1. A lot of general information from student textbooks. It is totally confusing to read for example the section «snRNA and HCC» starting with a description of siRNA (which are not expressed in mammals and have nothing with HCC if we talk about endogenous molecules) and snoRNA which are in fact housekeeping RNA.  Moreover, miRNA, siRNA and snoRNA are far not the only classes of small ncRNA.
  2. -«linear lncRNAs are longer than 200 nucleotides and share several common features with mRNAs, such as the transcription by RNA polymerase II, the 5’capping, the splicing, and the polyadenylation»Not truth, as not all of the lncRNAs undergo processing especially polyadenylation
  3. there is absolutely no need to list «five categories» on lncRNA in details (intergenic, antisense, enhancer…) as it is again textbook information with no relation to the present review
  4. The authors use inappropriate phrases like «RNA non-coding genetic sequences» or «non-coding genetic sequences (ncRNAs)». NcRNAs are transcripts, not genetic sequences. Yeap, they are encoded in the genome by their genes, but they act as RNA molecules
  5. There are a lot of recent as well as previous reviews concerning lncRNAs and their role in HCC. The authors should highlight what is the main difference and focus of their review that bring it novelty
  6. The selection of exact examples on miRNAs and lncRNA is not clear.  for example, in the section «Role of microRNAs dysregulation in hepatocellular carcinoma progression»

    the authors just took one individual publication 10.1038/s41598-020-71324-z 
    and listed 4 miRNA from this article one by one with detailed description. 

However there are much more known miRNAs even if we take into consideration only HCC progression:

Oura K, Morishita A, Masaki T. Molecular and Functional Roles of MicroRNAs in the Progression of Hepatocellular Carcinoma-A Review. Int J Mol Sci. 2020 Nov 7;21(21):8362. doi: 10.3390/ijms21218362.

Mizuguchi Y, Takizawa T, Yoshida H, Uchida E. Dysregulated miRNA in progression of hepatocellular carcinoma: A systematic review. Hepatol Res. 2016 Mar;46(5):391-406. doi: 10.1111/hepr.12606. 

as well as general detailed reviews such as 

Koustas E et al An Insight into the Arising Role of MicroRNAs in Hepatocellular Carcinoma: Future Diagnostic and Therapeutic Approaches. Int J Mol Sci. 2023 doi: 10.3390/ijms24087168.
Li J, Bao H, Huang Z, Liang Z, Wang M, Lin N, Ni C, Xu Y. Little things with significant impact: miRNAs in hepatocellular carcinoma. Front Oncol. 2023 May 19;13:1191070. doi: 10.3389/fonc.2023.1191070. 

Comments on the Quality of English Language

English is appropriate with some minor typos and etc

Reviewer 2 Report

Comments and Suggestions for Authors

Very good review! Although significant progress has been made in the study of liver cancer, there are still many unknown areas. Non-coding RNAs (ncRNAs) is a research hotspot.The authors describe the main dysregulated sncRNAs and lncRNAs and the relative molecular pathways involved in HCC progression, analyzing their specific implications in certain etiologically-related contexts, with a concomitant overview of their usefulness and applicability in clinical practice as novel diagnostic and prognostic tools.Therefore, after reviewing the manuscript carefully, I think this study should be accepted for publication with minor revision. 

   Minor concern

   There are some minor formatting issues in parts of the article that the author should improve. For example,the length of the abstract is too long, so it should be modified appropriately

Reviewer 3 Report

Comments and Suggestions for Authors

The authors present a narrative review on the "Role of non-coding RNAs in hepatocellular carcinoma progression: from classic to novel clinicopathogenetic implications". In this manuscript, the authors provide a thorough and detailed analysis of the contemporary literature of ncRNAs in HCC progression. By structuring their work in different subheadings, they achieve to give an overview of a wide topic, while I especially enjoyed the distinctive detailedness. Still, several points need to be addressed by the authors:

1. The plethora of ncRNAs may be confusing for the reader. I suggest some sort of Venn diagram, so that the reader may gain an overview of the different ncRNAs and their sub-RNAs.

2. Potential implications on the diagnosis and prognosis of patients with HCC by ncRNAs were greatly outlined in  the manuscript. However, I miss a paragraph on the potential implications on the therapy and management of patients with HCC. I think the manuscript would benefit tremendously.

3. Both figures have a low resolution. Please enhance.

4. In Figure 1 NAFLD is pictured as a sad liver with a burger and a beer in hands. I condemn this kind of connotation and encourage the authors to find a more suitable portrayal.

5. Throughout the manuscript, "gut" is spelled in capital letters. Why?

6. p4, l172: What does "LC" mean?

Reviewer 4 Report

Comments and Suggestions for Authors

This manuscript underscores the pivotal role of non-coding genetic sequences (ncRNAs) in the progression of hepatocellular carcinoma (HCC). Dysregulation of both small and long ncRNAs influences HCC-related molecular pathways, presenting a promising avenue for innovative diagnostic and prognostic approaches. Additionally, it delves into the interplay between ncRNAs, immune responses, oxidative stress regulation, and the composition of gut microbiota, revealing novel frontiers in our understanding of HCC progression. The review provides an extensive and nearly exhaustive account of the role of ncRNAs in HCC. The table is suitable, but enhancing it with corresponding references would further improve its quality. This manuscript can be considered for publication once it has addressed the comments provided by other reviewers and feedback from the editors

Comments on the Quality of English Language

Minor editing required